# Identification of Released Bacterial Extracellular Vesicles Containing Lpp20 from *Helicobacter pylori*

**DOI:** 10.3390/microorganisms13040753

**Published:** 2025-03-26

**Authors:** Aoi Okamoto, Tatsuki Shibuta, Nanaka Morita, Ryota Fujinuma, Masaya Shiraishi, Reimi Matsuda, Mayu Okada, Satoe Watanabe, Tsukuru Umemura, Hiroaki Takeuchi

**Affiliations:** 1Medical Laboratory Science, Graduate School of Health and Welfare Sciences, International University of Health and Welfare, 4-3 Kozunomori, Narita 286-8686, Japan; 24s3068@g.iuhw.ac.jp (A.O.); 23s1089@g.iuhw.ac.jp (M.S.); 24s3053@g.iuhw.ac.jp (R.M.); 24s1026@g.iuhw.ac.jp (M.O.); 2Department of Medical Science Technology, School of Health Science at Fukuoka, International University of Health and Welfare, 137-1 Enokiz, Okawa 831-8501, Japan; tshibuta@iuhw.ac.jp (T.S.); umemura@iuhw.ac.jp (T.U.); 3Department of Medical Science Technology, School of Health Science at Narita, International University of Health and Welfare, 4-3 Kozunomori, Narita 286-8686, Japan; nmorita@iuhw.ac.jp (N.M.); 2157075@g.iuhw.ac.jp (R.F.); 2157086@g.iuhw.ac.jp (S.W.)

**Keywords:** *Helicobacter pylori* (*H. pylori*), bacterial extracellular vesicles (bEVs), Lpp20, extragastric disease

## Abstract

*Helicobacter pylori* is a pathogenic bacterium that causes gastric and extragastric diseases. We have previously demonstrated that one of the mechanisms of *H. pylori*-associated chronic immune thrombocytopenia involves immune complexes of platelets, a *H. pylori* protein Lpp20 and an anti-Lpp20 antibody. However, it remains unclear how Lpp20 enters the body. We hypothesize that bacterial extracellular vesicles (bEVs) transport Lpp20. Thus, this study assessed Lpp20 in the bEVs released from seven clinical *H. pylori* isolates, using immunoprecipitation (IP), immunoblotting (IB), and surface plasmon resonance imaging (SPRi), with anti-GroEL (a marker of bEVs) and anti-Lpp20 antibodies. Lpp20 and bEVs were each detected in lysates of all seven strains. IP–IB experiments demonstrated that bEVs containing Lpp20 were produced by five of the strains (J99, SS1, HPK5, JSHR3, and JSHR31). SPRi using an anti-Lpp20 antibody demonstrated significantly higher reflectance from the strain HPK5 than from its *lpp20*-disrupted strains (*p* < 0.01), indicating localization of Lpp20 on the bEVs’ surface; Lpp20 may also be contained within bEVs. The bEVs containing Lpp20 were not detected from two clinical *H. pylori* strains (26695 and JSHR6) or from two *lpp20*-disrupted strains (26695ΔLpp20 and HPK5ΔLpp20). Differences in Lpp20 detection in bEVs are likely due to variations in bEV production resulting from strain diversity.

## 1. Introduction

*Helicobacter pylori* is a Gram-negative spiral-shaped pathogenic bacterium that was first reported in 1983 [1]. Many studies have shown that *H. pylori* infection is associated not only with upper gastrointestinal diseases [2] but also with various extragastric diseases, including autoimmune [3], hematological [4], cardiovascular [5], neurologic [6], skin [7], metabolic-related [8], hepatobiliary [9], and eye diseases [10]. Extragastric diseases caused by *H. pylori* infection can be improved by *H. pylori* eradication, but with geographical/regional differences [11]. In Japan, improvement in platelet counts has been reported in >50% of *H. pylori*-infected patients with chronic immune thrombocytopenia (cITP) [12,13,14]. We previously reported that one of the mechanisms underlying the development of *H. pylori*-associated cITP is that the *H. pylori* outer membrane protein Lpp20 binds to platelets, forms an immune complex with anti-Lpp20 antibodies, and induces platelet destruction and thrombocytopenia [15]. Furthermore, platelets bound to Lpp20 are aggregated and activated, suggesting a role of this protein in the development of other extragastric diseases such as thrombosis-mediated acute coronary syndrome (ACS) and chronic urticaria [15,16,17]. However, there is no case report to date of bacteremia caused by *H. pylori*. It is unclear how the bacterial component Lpp20 is transferred from the stomach into the bloodstream to contribute to the development of *H. pylori*-associated extragastric diseases.

Extracellular vesicles (EVs) released by various types of cells have been a rapidly expanding focus of research in recent years. However, the classification and nomenclature of EVs derived from bacteria lack consistency, highlighting the need for standardization. Thus, in accordance with guidelines issued by the International Society for Extracellular Vesicles (ISEV), EVs produced and released by bacteria are now collectively referred to as bacterial EVs (bEVs) [18]. bEVs released from bacteria in animals have demonstrated that bEVs transport bacterial components throughout the body. The bEVs are lipid bilayer vesicles with a diameter of 20–300 nm that contain DNA, RNA, proteins, and other components from the parent bacteria. They are mainly secreted during the growth stage [19,20,21,22]. The bEVs serve as carriers that protect their molecular cargoes from the external environment. Interestingly, bEVs have double-sided properties—an aggressive aspect, delivering virulence factors to host cells, and a defensive aspect, mediating interbacterial communication via quorum sensing [22,23,24,25,26,27,28], whose features function to protect the bacteria. *H. pylori* bEVs were first analyzed by double silver staining in 1997 [29], and it has been shown that *H. pylori* bEVs containing vacuolating cytotoxin (VacA) are delivered to gastric epithelial cells [21]. High-performance liquid chromatography–mass spectrometry (HPLC–MS/MS) analysis of two strains showed that *H. pylori* bEVs contain >400 molecules, including Lpp20, and that the contents of bEVs are strain-dependent [20,28]. However, there is no direct evidence regarding identification of bEVs containing Lpp20. Thus, this study was performed using seven clinical *H. pylori* strains isolated in geographically different regions; immunoblotting (IB), immunoprecipitation (IP), and surface plasmon resonance imaging (SPRi) analysis were used to assess Lpp20 in *H. pylori* bEVs.

## 2. Materials and Methods

### 2.1. Bacterial Strains and Growth Conditions

In this study, we used *Escherichia coli* (*E. coli*) and nine *H. pylori* strains (Table 1). The *H. pylori* strains included seven clinical isolates with diverse geographical origins (strains 26695/ATCC700392, J99/ATCC700824, SS1/ATCC43504, HPK5, JSHR3, JSHR6, and JSHR31), and two *lpp20* gene-disrupted strains (26695ΔLpp20 and HPK5ΔLpp20), derived from strains 26695 and HPK5, respectively, generated in our laboratory. The whole genome sequences of three isolates from Japan (JSHR3, JSHR6, and JSHR31) have been registered [30], and they are recommended as standard strains for susceptibility testing in Japan.

*H. pylori* was cultured in *Brucella* broth (BB; Becton Dickinson, Franklin Lakes, NJ, USA) supplemented with 10% horse serum (HS) and 10 μg/mL vancomycin (Sawai Pharmaceutical, Osaka, Japan). This medium is referred to as BB-liquid medium (BBL); BBL supplemented with 1.4% agar was used to prepare agar plates (BB plates). *H. pylori* were cultured at 37 °C under 10% CO_2_ for 72 h according to a previous report [15]. The bacterial strains and plasmids (for *lpp20* gene disruption) used in this study are shown in Table 1.

### 2.2. Generation of lpp20 Gene (HP1456) Disruption Strains

The *lpp20*-disrupted strains were prepared using constructed plasmids and homologous recombination, according to a previous report [35]. Briefly, the genomic DNAs of *H. pylori* strains 26695 and HPK5 were used for the amplification of the full-length *lpp20* gene (528 bp) and its flanking open reading frames (HP1455 and HP1457) by PCR (Table 2). The resulting 1.3 kb PCR product was cloned into the pGEM-Teasy vector (Promega, Madison, WI, USA), yielding plasmids containing the *lpp20* gene (p*lpp20*E-1 and p*lpp20*E-2, respectively). Next, these plasmids (“the p*lpp20*E plasmids”) were digested with *Bam*HI (New England Biolabs, Ipswich, MA, USA), a cleavage site for what is within the *lpp20* gene. The 1.3 kb kanamycin resistance gene (*kan*^r^) obtained from *Bam*HI-digested pUK4k was ligated into the *Bam*HI-digested p*lpp20*E plasmids using T4 DNA ligase (Promega, USA), followed by plasmid transformation into *E. coli* DH5α. As a result, the transformants selectively grew on Luria–Bertani broth (Nacalai Tesque, Kyoto, Japan)-agar plates containing 10 μg/mL kanamycin and yielded the constructed plasmids p*lpp20*E-km-1 and p*lpp20*E-km-2 in which the *H. pylori lpp20* gene was disrupted by *kan*^r^. The p*lpp20*E-km plasmids were used for homologous recombination to construct *lpp20*-disrupted *H. pylori* strains 26695ΔLpp20 and HPK5ΔLpp20, which selectively grew on BB plate containing 10 μg/mL kanamycin. The *lpp20*-disrupted strains and the direction of the *Km*^r^ insert in the *lpp20* genes were confirmed by PCR with specific primers (Table 2). In addition, loss of Lpp20 was confirmed by IB with an anti-Lpp20 antibody generated in our laboratory.

### 2.3. Preparation of H. pylori Cell Lysates

*H. pylori* was cultured overnight in BBL at 37 °C under 10% CO_2_ with shaking at 100–115 rpm (Shaker SRR-2, AS ONE, Osaka, Japan). An aliquot of well-grown bacteria in BBL was subcultured in fresh BBL overnight. This process was repeated twice to more precisely match the growth conditions of individual bacteria. Finally, the bacterial culture was adjusted to OD_600 nm_ = 0.3 in fresh BBL, incubated overnight, and collected. The culture was centrifuged at 7000× *g* for 30 min at 4 °C to separate the bacterial cells and the supernatant. The supernatant was processed for bEV preparation as described in Section 2.4. The bacterial cells were suspended in 400 μL of phosphate-buffered saline (PBS), homogenized on ice using an Omni tissue homogenizer (Omni, Dallas, TX, USA), and centrifuged at 12,000× *g* for 3 min at 4 °C. We used the supernatant as the *H. pylori* lysate. The protein concentration of lysates was measured using a Protein Assay Rapid Kit Wako II (Fujifilm Wako, Osaka, Japan). The lysate was stored at −20 °C until use.

### 2.4. Preparation of H. pylori bEVs

*H. pylori* bEVs were prepared according to previously described methods [36,37,38], with minor modifications. Briefly, the culture supernatant collected as described in Section 2.3 was further centrifuged at 12,000× *g* for 45 min at 4 °C to eliminate residual bacterial cells and debris. Subsequently, the cleared supernatant was passed through 0.45 and 0.22 μm filter membranes to remove other large molecules, and the cleared sample was prepared. The sample prepared was used in this study as *H. pylori* bEVs; bEVs were stored at −20 °C until use.

### 2.5. Visualization and Quantitative Analysis of bEVs’ Size and Concentration

We performed transmission electron microscopy (TEM) and nanoparticle tracking analysis (NTA) according to the ISEV’s recommended methods [18] to identify bEVs in the prepared samples. TEM analysis was performed by JEOL JEM-1400Flash electron microscopy at 100 kV at the Hanaichi UltraStructure Research Institute (Japan; https://www.kenbikyo.com/ (accessed on 17 February 2025)). Briefly, a droplet of prepared sample was placed on a carbon-film grid for 10 s. Staining solution, 2% uranyl acetate in water, was added to the grid and allowed to stain for 10 s, then the dried grid was subjected to electron microscopic observation with negative staining. NTA was performed using a NanoSight analyzer by Fujifilm Wako (Japan; https://labchem-wako.fujifilm.com/jp/custom_service/products/95163.html (accessed on 6 March 2025)). Briefly, bEV samples prepared from the strains HPK5 and HPK5ΔLpp20 were diluted in ultrapure water (Milli-Q, Merck, Darmstadt, Germany), and Mili-Q water was used as a blank. NTA was performed in five 60 s reads. The average of the five reads was calculated and plotted as particle size versus number of particles per milliliter. The recordings were processed by NTA 2.3 build 0033 software using a detection threshold of 8.

### 2.6. Immunoprecipitation

To analyze the properties of bEVs prepared from nine *H. pylori* strains, IP using anti-rabbit IgG-coated-magnetic beads (Thermo Fisher Scientific, Waltham, MA, USA) was conducted according to the manufacturer’s instructions. Briefly, each primary antibody (an anti-GroEL antibody by Sigma-Aldrich or the anti-Lpp20 antibody prepared in our laboratory [15]) was added to a bEV suspension and incubated at 4 °C for 1 h. Then, anti-rabbit IgG-coated magnetic beads washed with a buffer [0.1% bovine serum albumin (BSA)–PBS containing 2 mM ethylenediaminetetraacetic acid (EDTA)] were added into the mixtures and incubated at 4 °C overnight. After the reaction, the anti-rabbit IgG-coated-magnetic beads were washed with buffer (0.1% BSA–PBS containing 2 mM EDTA and 0.1% Tween 20), suspended in 1× sodium dodecyl sulfate (SDS) buffer and then heated at 90 °C for 5 min. The magnetic beads were removed using a magnet, and the resultant suspensions were stored at −20 °C until use. The bEV suspensions were subjected to IP–IB. IP without primary antibodies was performed as a negative control. In this study, we performed immunoprecipitation (IP) and immunoblotting (IB) to investigate whether Lpp20 is present on the surface and/or inside of bEVs. IP with an anti-Lpp20 antibody specifically captured Lpp20, and the precipitates were subjected to IB with an anti-GroEL antibody to detect bEVs, providing evidence that Lpp20 is localized on the surface of bEVs. Furthermore, we performed IP–IB with swapped antibodies (IP with an anti-GroEL antibody and IB with an anti-Lpp20 antibody) to confirm the presence of bEVs containing Lpp20.

### 2.7. Immunoblotting

The bacterial lysates and bEV suspensions treated without and bEV suspensions not subjected to IP, and bEV suspensions treated by IP were subjected to IB according to a previous report [15]. Briefly, samples mixed in an equal volume of 2× SDS buffer were heated at 90 °C for 5 min and then subjected to SDS–PAGE using an XV PANTERA GEL MP system (15% and 5–20%; DRC, Osaka, Japan). The proteins were transferred onto polyvinylidene fluoride membranes (Merck Millipore, Burlington, MA, USA) by using a MINICA-MP blotting module with Tris-glycine buffer (43 V, 30 min). The membrane was blocked with 0.5% skim milk and reacted with the primary antibody and then the secondary antibody. Finally, the membrane was treated with a Pierce ECL Plus Western Blotting Substrate (Thermo Fisher Scientific, USA), and bands were visualized by using an LAS-4000mini Luminescent Image Analyzer (Fujifilm Life Science, Tokyo, Japan). We used two primary antibodies: an anti-GroEL antibody for detection of BEVs [39,40,41] and an anti-Lpp20 antibody for detection of *H. pylori* Lpp20 [15]. As the secondary antibody, we used horseradish peroxidase-labeled goat anti-rabbit IgG (Abcam, Cambridge, UK).

### 2.8. Surface Plasmon Resonance Imaging (SPRi) Analysis

To evaluate the binding interactions of bEVs, SPRi analysis was performed using the OpenPleX system (HORIBA, Kyoto, Japan). First, 1 mL of a miRCURY Exosome Cell/Urine/CSF Kit (Qiagen, Venlo, The Netherlands) was added to 4 mL of the bEV suspensions of four samples (from strains HPK5, HPK5ΔLpp20, 26695, and 26695ΔLpp20), followed by vortexing and incubation at 4 °C overnight. The samples were centrifuged at 3000× *g* for 30 min at 25 °C to collect the bEVs as a pellet. The supernatant was discarded, and the pellet was washed twice with PBS with centrifugation at 3000× *g* for 5 min at 25 °C. After removing the supernatant, the EV pellet was resuspended in 0.1% casein in Dulbecco’s phosphate-buffered saline without calcium and magnesium [D-PBS (−); Fujifilm Wako, Japan], then diluted 50–200-fold and used for SPRi analysis. The biochip had a gold film activated through esterification, and when antibodies (anti-GroEL and anti-Lpp20 antibodies) were dropped onto the gold surface, they were immobilized on the surface through an amine coupling reaction. An antibody-bound biochip was incubated overnight with 1% casein–PBS solution for blocking. The prepared bEV samples were introduced into the SPRi system at a flow rate of 25 μL/min at 25 °C for 8 min. After each measurement, the sensor surface was regenerated with 10 mM glycine–HCl (pH 2.5) to remove bound bEVs. Data analysis was performed using Scrubber software version 2.0g (BioLogic Software, Canberra, Australia). A blank spot served as a negative control, and the signal intensities for analysis were calculated by subtracting the average signal from the blank spot. SPRi analysis for detection of bacterial-derived bEVs enhances the potential for future clinical applications.

## 3. Results

### 3.1. Identification and Characterization of bEVs of the Samples Used in This Study

We observed the ultrastructural morphology of *H. pylori* bEVs by TEM (Figure 1a). NTA revealed that the mean particle diameter of the bEVs from strains HPK5 and HPK5ΔLpp20 was 142.47 ± 8.91 nm and 139.34 ± 2.11 nm, respectively. The bEV concentration was (6.26 ± 1.78) × 10^9^ particles/mL for the strain HPK5 and (4.30 ± 0.19) × 10^9^ particles/mL for the strain HPK5ΔLpp20 (Figure 1b,c). An average of 2–3 bEVs per field (×10,000) was observed in the samples. These findings confirmed the presence of bEVs in the samples prepared for use in this study. Furthermore, no significant differences in particle size or concentration were observed between the strain HPK5 and its *lpp20*-disrupted derivative strain, HPK5ΔLpp20.

### 3.2. Analysis of Lpp20 in Cell Lysates and bEVs from Seven Clinical H. pylori Strains and Two lpp20-Disrupted Strains

By IB with an anti-Lpp20 antibody, Lpp20 was detected in the cell lysates of seven clinical *H. pylori* strains (26695, J99, SS1, HPK5, JSHR3, JSHR6, and JSHR31) (Figure 2). The bEVs were detected by IB with an anti-GroEL antibody for all seven clinical strains without IP (Figure 3).

Next, bEV suspensions without IP were subjected to IB with an anti-Lpp20 antibody. bEVs containing Lpp20 were detected from five strains (J99, SS1, HPK5, JSHR3, and JSHR31), but not from the strains 26695 or JSHR6 (Figure 3).

Lpp20 was not detected in the cell lysate or bEVs from the two *lpp20*-disrupted strains (26695ΔLpp20 and HPK5ΔLpp20); these strains released bEVs (Figure 2 and Figure 3).

### 3.3. IP–IB Analysis

We performed IP–IB analysis to confirm the bEVs containing Lpp20 released from *H. pylori* clinical strains J99, SS1, HPK5, JSHR3, and JSHR31. IP (anti-Lpp20 antibody)–IB (anti-GroEL antibody) analysis enabled detection of bEVs from the five strains (Figure 4). Next, we performed IP (anti-GroEL antibody)–IB (anti-Lpp20 antibody) analysis to confirm the presence of Lpp20 in the bEVs from these five strains. Lpp20 was not detected in bEVs from the strain HPK5ΔLpp20, in which the *lpp20* gene was disrupted (Figure 4). These data are consistent with the results obtained using bEV suspensions without IP (Section 3.2). The bEVs contained surface-localized Lpp20; this protein may also be contained within bEVs.

### 3.4. SPRi Analysis

In SPRi analysis with an anti-Lpp20 antibody, significantly more spots were detected for bEVs of the strain HPK5 than for the strain HPK5ΔLpp20 (*p* < 0.01). Minor signals were observed in a BBL medium including 10% HS (BBL + HS, no bacteria), which was used as the blank (Figure 5a); thus, the results were adjusted on the basis of the reflectivity of the blank to evaluate the binding interactions of bEVs (Figure 5b). No signal was observed in the BBL medium.

In the results from SPRi analysis with an anti-GroEL antibody, numerous spots were detected for the bEVs of both strains HPK5 and HPK5ΔLpp20 without a significant difference (Figure 5a). The binding interactions of bEVs were evaluated in a similar manner as described above (Figure 5b).

## 4. Discussion

*H. pylori* infection is thought to be related to gastric and extragastric disorders such as cITP and ACS, but these have geographically different occurrence rates [3,11]. The bEVs released from bacteria, including *H. pylori*, are suggested to be involved in the delivery of bacterial molecules/pathogens throughout the body [20,21,28]. Here, we investigated *H. pylori* bEVs containing Lpp20 in seven clinical strains originating from geographically different areas of the world. *H. pylori* Lpp20 is associated with cITP and thrombosis-mediated ACS. In this study, Lpp20 was expressed in and bEVs were released by all seven clinical strains (26695, J99, SS1, HPK5, JSHR3, JSHR6, and JSHR31) irrespective of their geographic origin. However, bEVs containing Lpp20 were detected from only five of the strains (HPK5, JSHR3, and JSHR31 isolated in Japan, J99 from the United States, and SS1 from Australia), but not from the strains 26695 (isolated in the United Kingdom) and JSHR6 (from Japan). These data show a difference in the content of bEVs generated and released from individual *H. pylori* strains, indicating different mechanisms of EV packaging between strains. The different properties of bEVs containing Lpp20 may influence the development of cITP associated with *H. pylori* infection. In fact, the effectiveness of *H. pylori* eradication therapy for cITP varies depending on the geographic region: In East Asia, including Japan, as well as Central and South America and Italy, efficacy rates are ≥50%, whereas the rate is <20% elsewhere in Europe (the United Kingdom, France, and Spain) and the United States [42,43,44,45]. In this study, bEVs containing Lpp20 were not detected from the strains 26695 and JSHR6, even though these strains expressed Lpp20 within the cells. At least, such diversity of bEVs may influence the development of various extragastric diseases associated with *H. pylori*.

IP–IB analysis confirmed the properties of the *H. pylori* bEVs containing Lpp20 described above. Lpp20 was present on the surface of the bEVs from the strains HPK5, JSHR3, JSHR31, J99, and SS1, and it may also be contained within the bEVs from these strains. Comprehensive analysis by HPLC–MS/MS of two strains (26695 and 11637) showed that *H. pylori* bEVs contain many molecules including Lpp20 and that the bEV content varies between strains [20,28]. Interestingly, bEVs containing Lpp20 could not be detected from 26695 in the present study, probably because of the non-ultracentrifugation-based bEVs preparation method we used. However, bEVs collected by ultracentrifugation may include other molecules and bacterial cell debris. Proteomics analysis of *H. pylori* bEVs collected by ultracentrifugation revealed that the most abundant protein among the unique peptides was the 60-kDa chaperonin GroEL [40]. In this study, we confirmed bEV existence in the samples prepared without ultracentrifugation and detected *H. pylori* bEVs using an anti-GroEL antibody. Given the variation of bEVs, more investigations with other antibodies, such as anti-outer membrane proteins and anti-urease antibodies, are needed to understand the individual properties of *H. pylori* bEVs. Furthermore, *H. pylori* possesses high genetic diversity, which enables it to persistently infect the stomach and adapt to changing conditions and environmental stresses [46]. It is necessary to clarify how the environmental conditions reflect and affect bacterial behaviors including the contents of released bEVs. Nevertheless, we directly identified bEVs containing Lpp20 and found different contents of bEVs among the seven clinical *H. pylori* strains that we tested, consistent with the results of earlier HPLC–MS/MS analyses.

In SPRi analysis using an anti-Lpp20 antibody, the binding interactions of bEVs meant that the reflectance ratio was significantly higher for the strain HPK5 than for the strain HPK5ΔLpp20 (*p* < 0.01). Meanwhile, the reflectance ratio using an anti-GroEL antibody-bound biochip did not show a significant difference between the two strains. These data indicated that the loss of Lpp20 did not significantly affect the production and release of bEVs from the strain HPK5ΔLpp20. Furthermore, consistent with the results from IP–IB, the SPRi data showed that Lpp20 was present at least on the surface of the released bEVs. However, the reflectivity observed using the anti-GroEL antibody-bound biochip did appear to be lower for the strain HPK5ΔLpp20 than for the strain HPK5. It cannot be excluded that Lpp20, an outer membrane protein of *H. pylori*, influences the bEVs production process in this species. It is necessary to elucidate the molecular function of Lpp20 in *H. pylori*.

To our knowledge, bEVs from bacteria (including *H. pylori*) have not been detected in blood or body fluid. Nevertheless, we consider it a possibility that bEVs deliver pathogenic factors throughout the body via the bloodstream. Released bEVs are taken up by host cells such as gastric epithelial cells [21,47], and these cells eventually produce exosomes including bEV contents and deliver them to extragastric tissues via the bloodstream. The CagA protein, encoded by *cytotoxin-associated gene A* (*cagA*), a pathogenic factor specific to *H. pylori*, has been strongly implicated in the development of gastric cancer [4]. The bEVs containing CagA were detected in the blood of patients with *CagA*-positive *H. pylori* infection [4,48]. Thus, we speculate that bEVs containing Lpp20 are involved in extragastric diseases. A further investigation into the properties of bEVs and/or bEVs-related transport systems in humans is important to elucidate the pathophysiology of extragastric *H. pylori*-related diseases and for the development of liquid biopsies for clinical practice. We are planning clinical trials using IB and SPRi analysis to address these issues.

## 5. Conclusions

In this study, we assessed the properties of *H. pylori* bEVs using seven clinical isolates from geographically distinct regions and laboratory-generated *lpp20*-disrupted strains. Lpp20 and bEVs were detected for all seven clinical strains. bEVs containing Lpp20 were detected from five of the seven clinical strains. Lpp20 was found on the surface of bEVs and may also be contained within bEVs. The results suggest that the mechanism by which proteins are processed and packed into bEVs depends on the individual *H. pylori* strain. By elucidating the strain diversity affecting bEV production, we could better understand the discrepancy between *H. pylori* infection rates and disease onset rates, paving the way for clinical trials. Further research is needed to elucidate the bEV packaging and transport systems for understanding the pathophysiology of extragastric diseases associated with *H. pylori*.

## Figures and Tables

**Figure 1 microorganisms-13-00753-f001:**
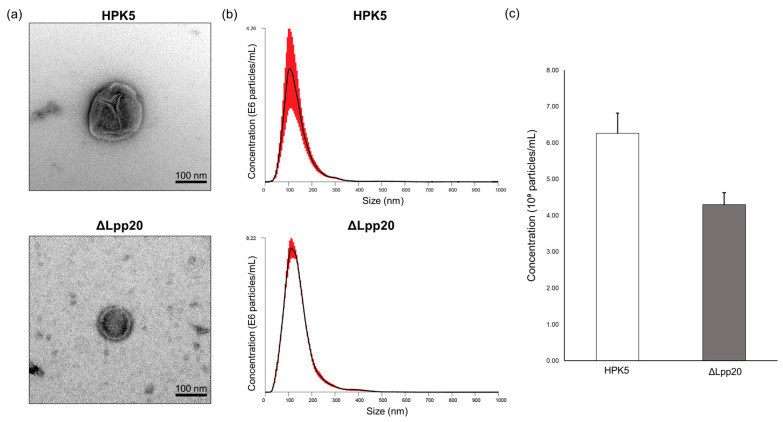
The characterization of bEVs from HPK5 and HPK5ΔLpp20 strains. (**a**) TEM image (×50,000) of bEVs derived from HPK5 (upper) and HPK5ΔLpp20 (lower) (scale bar = 100 nm); (**b**) NTA shows the average concentration of bEVs (particles/mL) of a particular size (nm) for HPK5 (upper) and HPK5ΔLpp20 (lower). The black peak represents the mean values from five independent measurements (*n* = 5). The red lines on the peaks indicate the standard error of the mean ± S.E.M.; (**c**) The graph illustrates the particle concentrations (10^9^ particles/mL) of HPK5 and HPK5ΔLpp20, as determined by NTA (mean ± S.E.M., *n* = 5). HPK5, a clinical strain in Japan (wild type); ΔLpp20, an HPK5-derived *lpp20*-disrupted strain.

**Figure 2 microorganisms-13-00753-f002:**
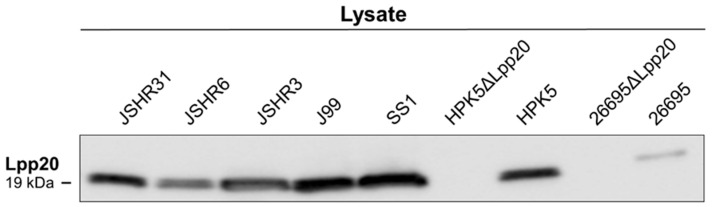
Analysis of Lpp20 in cell lysates from seven clinical *H. pylori* strains and two *lpp20*-disrupted strains (HPK5ΔLpp20 and 26695ΔLpp20). By immunoblotting (IB) with an anti-Lpp20 antibody, Lpp20 was detected in all seven *H. pylori* lysates (JSHR31, JSHR6, JSHR3, J99, SS1, HPK5, and 26695). Lpp20 was not detected in two *lpp20*-disrupted strains (HPK5ΔLpp20 and 26695ΔLpp20).

**Figure 3 microorganisms-13-00753-f003:**
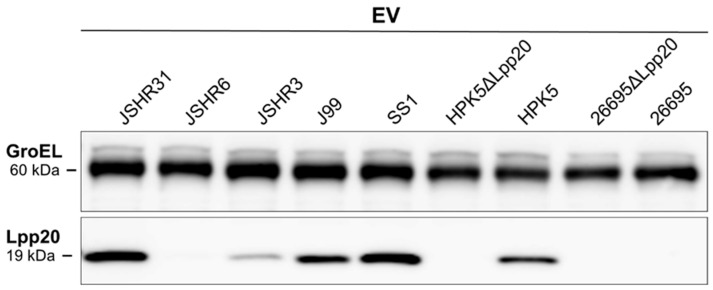
Analysis of Lpp20 in bEVs from seven clinical *H. pylori* strains and two *lpp20*-disrupted strains. The bEVs were detected by IB with an anti-GroEL antibody in all nine bEV suspensions without IP. Next, the bEV suspensions without IP were subjected to IB with an anti-Lpp20 antibody, demonstrating that bEVs containing Lpp20 were detected in five strains (JSHR31, JSHR3, J99, SS1, and HPK5). The bEVs containing Lpp20 were not detected in other two bEV suspensions (from the strains JSHR6 and 26695). Lpp20 was not detected in bEVs from HPK5ΔLpp20 and 26695ΔLpp20 strains that release bEVs.

**Figure 4 microorganisms-13-00753-f004:**
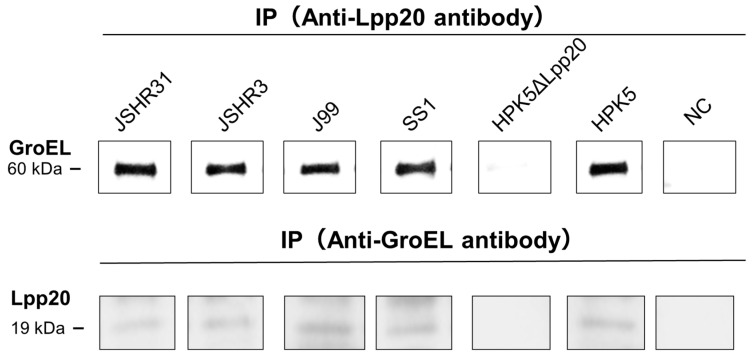
Confirmation of bEVs containing Lpp20 by IP–IB. “NC” denotes negative control without a primary antibody, the HPK5ΔLpp20 strain represents the *lpp20*-disrupted strain derived from *H. pylori* HPK5. The upper panel shows the detection of bEVs (60 kDa) by IB (Anti-GroEL antibody) followed by IP (Anti-Lpp20 antibody). The lower panel shows the detection of Lpp20 (19 kDa) by IB (Anti-Lpp20 antibody) followed by IP (Anti-GroEL antibody).

**Figure 5 microorganisms-13-00753-f005:**
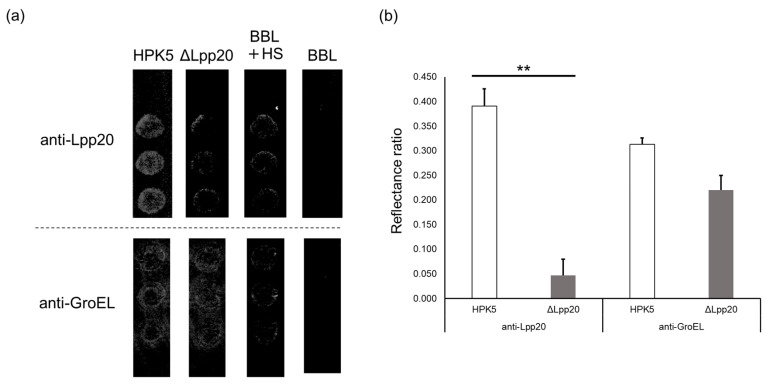
Evaluation of binding interaction of bEVs by SPRi analysis. (**a**) SPRi analysis is performed in triplicate, and the signals are shown. The bEVs signals were detected using a biochip immobilized with an anti-Lpp20 antibody (upper) and an anti-GroEL antibody (lower); (**b**) On one hand, the evaluation for the binding interactions of bEVs. The reflectance ratio is significantly higher in the strain HPK5 than in the strain HPK5ΔLpp20 (**, *p* < 0.01) using an anti-Lpp20 antibody. On the other hand, the reflectance ratio is even in two strains using an anti-GroEL antibody. HPK5, the strain HPK5; ΔLpp20, the *lpp20*-disrupted strain HPK5; BBL + HS, a *Brucella* broth liquid medium including 10% horse serum; BBL, *Brucella* broth liquid medium only.

**Table 1 microorganisms-13-00753-t001:** Bacterial strains and plasmids.

Bacterial Strain or Plasmid	Genotype or Characteristics ^α^	Region	Reference
*H. pylori*			
26695	Wild type	the United Kingdom	[31]
J99	Wild type	the United States	[32]
SS1	Wild type	Australia	[33]
HPK5	Wild type	Japan	[34]
JSHR3	Wild type	Japan	[30]
JSHR6	Wild type	Japan	[30]
JSHR31	Wild type	Japan	[30]
26695ΔLpp20	26695 derivative; *kan* in *lpp20*; Km^r^		This study
HPK5ΔLpp20	HPK5 derivative; *kan* in *lpp20*; Km^r^		This study
*E. coli*			
DH5α	F^−^ φ80*dlacZ*ΔM15 Δ(*argF-lac*)*U169 deoR recA1 endA1 hsdR17*(r_K_^−^ m_K_^+^) *supE44 thi-1 gyrA96 relA1*		GIBCO-BRL
Plasmids			
pGEM-Teasy	3.0 kb cloning vector; Ap^r^		Promega
p*lpp20*E-1	*lpp20* and its flanking ORFs (1.3 kb) of 26695 in pGEM-Teasy; Ap^r^		This study
p*lpp20*E-2	*lpp20* and its flanking ORFs (1.3 kb) of HPK5 in pGEM-Teasy; Ap^r^		This study
p*lpp20*E-*km*-1	1.3 kb *kan* in *lpp20* of p*lpp20*E-1; Ap^r^, Km^r^		This study
p*lpp20*E-*km*-2	1.3 kb *kan* in *lpp20* of p*lpp20*E-2; Ap^r^, Km^r^		This study

^α^ *kan*, kanamycin resistance gene; ORF, open reading frame; Ap^r^, ampicillin resistance; Km^r^, kanamycin resistance.

**Table 2 microorganisms-13-00753-t002:** Primers used for *lpp20* gene disruption.

Target Genes	Primer Name	Sequence (5′-3′)
HP1455-HP1457	1457F-231	TTCAGATGTGATTAACGACACC
	1455R-390	CTCATTCATTAAAGCGACATGC
*lpp20*	1456F-241Bam	TTGGATCCCTACTAACCAAGCTACAGCG
	1456R-224	ATTAGTGATCAAATCTTCAGCC

F, forward primer; R, reverse primer.

## Data Availability

The original contributions presented in this study are included in the article. Further inquiries can be directed to the corresponding author.

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
