# Peer review of "Identification of Released Bacterial Extracellular Vesicles Containing Lpp20 from Helicobacter pylori"

_microorganisms, 2025, doi:10.3390/microorganisms13040753_

Round 1

Reviewer 1 Report

Comments and Suggestions for Authors

The manuscript presents a straightforward conclusion and plan, focusing on the identification of the Lpp20 protein in the extracellular vesicles of H. pylori bacteria, although it requires some significant corrections.

I suggest that the authors should further emphasize in the introduction section the hypothesis about the biological implications of the identification of this particular protein in bacterial extracellular vesicles in relation to the pathogenesis of H. pylori-related infections.

One of the major comments is that there is no evidence (except GroEL detection) that EVs are present in the obtained samples. The methodology section and the results should be supplemented with imaging of the isolated EVs (i.e. electron microscopy) and measurements of the size and concentration of EVs, using one of the methods recommended by ISEV, i.e. NTA, TRPS or DLS, as this is obligatory for studies based on EVs to standardize EVs research and ensure reproducibility

Additionally, there should be included a more detailed description of the methodology of isolation of bacterial extracellular vesicles. Can the authors exclude contamination of the preparation with soluble proteins in the applied method? Please address this issue.

Author Response

We showed the response to reviewers’ comments as below and revised manuscript according to these comments.

Comments and Suggestions for Authors

The manuscript presents a straightforward conclusion and plan, focusing on the identification of the Lpp20 protein in the extracellular vesicles of H. pylori bacteria, although it requires some significant corrections.

I suggest that the authors should further emphasize in the introduction section the hypothesis about the biological implications of the identification of this particular protein in bacterial extracellular vesicles in relation to the pathogenesis of H. pylori-related infections.

One of the major comments is that there is no evidence (except GroEL detection) that EVs are present in the obtained samples. The methodology section and the results should be supplemented with imaging of the isolated EVs (i.e. electron microscopy) and measurements of the size and concentration of EVs, using one of the methods recommended by ISEV, i.e. NTA, TRPS or DLS, as this is obligatory for studies based on EVs to standardize EVs research and ensure reproducibility.

Additionally, there should be included a more detailed description of the methodology of isolation of bacterial extracellular vesicles. Can the authors exclude contamination of the preparation with soluble proteins in the applied method? Please address this issue.

→ Thank you for your comments. According to your comments, we performed TEM and NTA using the obtained samples and confirmed bEVs existence in the samples. We described more detail with figure (Fig. 1) in “2.4 and 2.5 of Materials and Methods” and “3.1 of Results” in the manuscript. Based on these results, it can be considered that the sample obtained through two rounds of centrifugation and two rounds of filtration is at least sufficiently purified for use in this analysis. Following ISEV recommendations, we have consistently used the term “bEVs” to refer to bacterial extracellular vesicles in this paper. We changed the title to “Identification of released bacterial extracellular vesicles containing Lpp20 from Helicobacter pylori” by adding “bacterial” in the title. We would appreciate your understanding.

Reviewer 2 Report

Comments and Suggestions for Authors

The manuscript entitled “Identification of Released Extracellular Vesicles Containing Lpp20 from Helicobacter pylori” give insights into the mechanism of H. pylori-associated immune thrombocytopenia by investigating the role of extracellular vesicles (EVs) in transporting Lpp20. The manuscript is generally clear, but the flow of information can be improved to enhance readability.

The methods are well-described; however the clarity could be improved by providing a brief explanation of why each technique was selected for detecting Lpp20 in EVs. It would also be interesting to understand why the authors chose clinical H. pylori isolates over ATCC strains. The conclusion mentions strain diversity affecting EV production, but it would be beneficial to suggest possible next steps or the broader implications of these findings. the study could benefit from further discussion regarding the limitations of strain diversity and how it might affect generalizability. In conclusion, I recommend including potential future research directions.

Minor remarks

Line 34 – omit in in the following sentence: …. bacterium that colonizes in the human stomach…

Line 39 – Please add the reference for: …but with geographical/regional differences

Line 41-43 – Please rephrase the sentence to enhance its English and clarity

Table 1 – add “Bacterial” before Strain or plasmid

Table 1 – correct Refarence to Reference

Line 229 – please add the reference for … but these have geographically different occurrence rates.

Comments on the Quality of English Language

English language should be improved to strength readability throughout the manuscript.

Author Response

We showed the response to reviewers’ comments as below and revised manuscript according to these comments.

Comments and Suggestions for Authors

The manuscript entitled “Identification of Released Extracellular Vesicles Containing Lpp20 from Helicobacter pylori” give insights into the mechanism of H. pylori-associated immune thrombocytopenia by investigating the role of extracellular vesicles (EVs) in transporting Lpp20. The manuscript is generally clear, but the flow of information can be improved to enhance readability.

The methods are well-described; however the clarity could be improved by providing a brief explanation of why each technique was selected for detecting Lpp20 in EVs. It would also be interesting to understand why the authors chose clinical H. pylori isolates over ATCC strains. The conclusion mentions strain diversity affecting EV production, but it would be beneficial to suggest possible next steps or the broader implications of these findings. the study could benefit from further discussion regarding the limitations of strain diversity and how it might affect generalizability. In conclusion, I recommend including potential future research directions.

→ Thank you for your comments. In this study, we performed immunoprecipitation (IP) and immunoblotting (IB) to investigate whether Lpp20 is present on the surface and/or inside bEVs. IP with an anti-Lpp20 antibody specifically captured Lpp20, and the precipitates were subjected to IB with an anti-GroEL antibody to detect bEVs, providing evidence that Lpp20 is localized on the surface of bEVs. Furthermore, we performed IP-IB with swapped antibody to confirm the presence of bEVs containing Lpp20. We incorporated these in “2.6 Immunoprecipitation”. (Line 165-170)

SPRi analysis for detection of the bacterial-derived EVs enhances the potential for future clinical applications. We incorporated these in “2.8 Surface plasmon resonance imaging (SPRi) analysis”. (Line 207-208)

Regarding to clinical H. pylori isolates over ATCC strains: according to your comments, we have given a brief explanation in appropriate position of “2.1 Bacterial strains and growth conditions” as follows: we used Escherichia coli (E. coli) and nine H. pylori strains, considering the geographical strain diversity: seven clinical isolates (26695/ATCC700392, J99/ATCC700824, SS1/ATCC43504, HPK5, JSHR3, JSHR6, and JSHR31), and two lpp20 gene-disrupted mutant strains (26695ΔLpp20 and HPK5ΔLpp20) derived from strains HPK5 and 26695 respectively, generated in our laboratory. Among four isolates from Japan, the whole genome sequences of three isolates (JSHR3, JSHR6, and JSHR31) have been registered (32), and they are recommended as standard strains for susceptibility testing in Japan (Table 1). (Line 77-84)

According to your comments, we incorporated the sentence in “5 Conclusions” as follows: By elucidating the strain diversity affecting bEV production, we could better understand the discrepancy between H. pylori infection rates and disease onset rates, paving the way for clinical trials. (Line 369-371)

Minor remarks

Line 34 – omit in in the following sentence: …. bacterium that colonizes in the human stomach… → Helicobacter pylori is a gram-negative spiral-shaped pathogenic bacterium that was first reported in 1983 [1].

Line 39 – Please add the reference for: …but with geographical/regional differences → we added the reference [11].

Line 41-43 – Please rephrase the sentence to enhance its English and clarity → We previously reported one of the mechanisms underlying the development of H. pylori-associated cITP as follows: the H. pylori outer membrane protein Lpp20 binds to platelets, forms an immune complex with anti-Lpp20 antibodies, and induces platelet destruction and thrombocytopenia [14]. 

Table 1 – add “Bacterial” before Strain or plasmid → We added “Bacterial” before Strain or plasmid in Table 1.

Table 1 – correct Refarence to Reference → We corrected.

Line 229 – please add the reference for … but these have geographically different occurrence rates. → we added the reference [3, 11]. (Line 296)

Round 2

Reviewer 1 Report

Comments and Suggestions for Authors

The authors have implemented several modifications in response to the prior comments provided. However, the visualization of EVs in a wider field of view is still lacking. And I still maintain the opinion that the EVs isolation methodology should be described in greater detail, and not only with references to previous publications.

Also, please change: '2.5  Proof of bEVs in the samples' to '2.5 Visualization and quantitative analysis of bEVs size and concentration'.

In addition, after changes, correction of spelling style and grammar is necessary.

Comments on the Quality of English Language

After the modifications, a thorough review and correction of spelling, style, and grammar is required.

Author Response

We showed the response to reviewer (R1) comments as below and revised manuscript according to the comments.

The authors have implemented several modifications in response to the prior comments provided. However, the visualization of EVs in a wider field of view is still lacking. And I still maintain the opinion that the EVs isolation methodology should be described in greater detail, and not only with references to previous publications.

Also, please change: '2.5  Proof of bEVs in the samples' to '2.5 Visualization and quantitative analysis of bEVs size and concentration'.

In addition, after changes, correction of spelling style and grammar is necessary.

→ Thank you for your comments.

We understood what you mean about the visualization of bEVs in the sample, and we outsourced and conducted two analyses such as TEM and NTA recommend, within the constraints of time and budget. The focus is the presence of bEVs in the samples used. Following your comments, we have confirmed this point using TEM and NTA analyses. Therefore, we believe there is no issue with the structure of the paper. By the way, we received a report indicating that, in the TEM analysis, an average of 2-3 bEVs per field of view (´10,000) was observed, whose sentence was added in the results. Furthermore, we added TEM image of HPK5ΔLpp20 strain (Fig.1) in the revised manuscript.  We would appreciate it if you could understand our situation with a limitation.  

According to your comment, we have provided a detailed description of the methods related to bEVs preparation in connection with cell lysate in “2.3 preparation of H. pylori cell lysate” and “2.4 preparation of H. pylori bEVs” as far as we can.  Furthermore, we changed to “2.5 Visualization and quantitative analysis of bEVs size and concentration” and revised the context, including information about analysis institution we commissioned. The revised paper has reviewed through a native English check.